# Leadless AV Pacemaker in Patient with Complete Heart Block and Bilaterally Implanted Two Deep Brain Stimulators Can Be Safe Therapeutic Option

**DOI:** 10.3390/ijerph20010388

**Published:** 2022-12-26

**Authors:** Jędrzej Michalik, Jonasz Kozielski, Mateusz Węclewicz, Roman Moroz, Maciej Sterliński, Marek Szołkiewicz

**Affiliations:** 1Department of Cardiology and Interventional Angiology, Kashubian Center for Heart and Vascular Diseases, Pomeranian Hospitals, 84-200 Wejherowo, Poland; 2Department of Neurosurgery, Copernicus Hospital, 80-803 Gdańsk, Poland; 31st Department of Heart Arrhythmia, National Institute of Cardiology, 04-628 Warsaw, Poland

**Keywords:** leadless cardiac pacemaker, deep brain stimulation, Parkinson’s disease

## Abstract

There are reports documenting that electromagnetic waves generated by deep brain stimulation devices can interfere with cardiac pacemakers. This might be even a life-threatening problem in cardiac pacemaker-dependent patients. Herein, we present a case report on a patient with bilaterally implanted deep brain stimulation devices, who concomitantly had the indications for permanent cardiac pacing. The report shows that a leadless AV pacemaker may be a safe and reasonable option in these cases.

## 1. Introduction

The introduction of transvenous cardiac pacemakers (PM) has brought a significant clinical benefit to patients with various heart rhythm problems, and the leadless devices are a step forward in cardiac electrotherapy development [1,2]. However, cardiac implantable pacemakers or cardioverter-defibrillators (CIED), are not the only pulse generators the patient may require. The concomitant use of CIED and deep brain stimulation (DBS) commonly used for the treatment of refractory Parkinson’s disease, essential tremor, and/or dystonia is not rare. The potential electromagnetic interference between the implanted devices can disturb their functioning and may lead to serious adverse consequences. There are some published reports on permanent cardiac pacing in DBS patients, but the data on the interactions between DBS and leadless cardiac pacemakers is highly limited [3]. 

## 2. Materials and Methods

This is a report on a patient with effective bilateral DBS due to medically refractory Parkinson’s disease and a symptomatic third-degree atrioventricular (AV) block with indications for permanent cardiac pacing. 

## 3. Case Report

A 74-year-old man, with a previously diagnosed severe aortic valve stenosis (aortic valve area ~0.6 cm^2^) and Parkinson’s disease, was referred to the Emergency Department due to syncope. The event occurred for the first time and it was not preceded by any other symptoms, although a general weakness persisted for a few weeks beforehand. An electrocardiogram (ECG) revealed a sinus rhythm with a third-degree atrioventricular block (Figure 1a), thus the patient was admitted to the Cardiology Department. 

Parkinson’s disease was diagnosed in 2000. The clinical effects of pharmacological treatment (Levodopa with Benserazide) were far from satisfactory, and moreover, after a few years of the response-based continuous drug dosage titration, this treatment also produced adverse reactions. In 2008, the patient underwent bilateral implantation of DBS (Medtronic Soletra 7426) with the electrodes placed in the ventrolateral part of the subthalamic nucleus (STN). The procedure was staged: first DBS pulse generator was implanted subcutaneously into the right subclavicular region, and a month later, the other one was implanted contralaterally. The DBS proved to be clinically efficient and the devices did not require any adjustments in the primarily accepted configuration (right DBS: 3.6 V, 60 µs, 130 Hz; left DBS: 3.5 V, 60 µs, 130 Hz; both devices set to monopolar modality). This allowed for the reduction of the daily Levodopa dosage significantly to control Parkinson’s disease symptoms and to maintain a good quality of life. The last replacement of DBS pulse generators due to battery depletion was performed in 2020 (Medtronic Activa SC 37602) and since that time, their configuration as well as a chronic pharmacological treatment have not been changed. 

At the time of admission to the Cardiology Department, the patient was in a good general condition. Echocardiography confirmed a severe aortic stenosis and revealed that his left ventricular ejection fraction was normal. The valve disease might also have been considered as a reason for syncope, however, the ECG clearly indicated that the patient primarily required a cardiac PM implantation as per the ESC 2021 guidelines class I recommendation [4]. We had a few options to minimize the risk of interferences, including the usage of epicardial leads, but after a discussion with a patient, we decided to implant a leadless AV pacemaker (LPM; Micra AV MC1AVR1, Medtronic). The LPM implantation was performed in the electrophysiology laboratory with the participation of a neurosurgeon, who turned off both DBS generators before the procedure. The LPM was implanted in the mid-septum of the right ventricle via the femoral vein access. The atrioventricular synchronous cardiac pacing was confirmed (Figure 1b) and the tests showed correct pacing threshold, ventricular sensing as well as impedance (0.25 V/0.24 ms; >20.0 mV; 920 ohms; respectively). Then, the DBS generators were turned back on and optimally programmed. We did not record any significant interference by DBS on LPM, although they were placed at a closer than recommended distance one from another (Figure 2). 

Nevertheless, we noticeably decreased the DBS pulse amplitude since it was suggested that a disruption in CIED functioning might be a function of increasing voltage and frequency in DBS [5]. Eventually, both DBS devices were set to bipolar modality and programmed as follow: right DBS—1.8 V, 60 µs, 130 Hz; left DBS—1.5 V, 60 µs, 130 Hz. Despite the lower voltage of bipolar stimulation, the patient did not aggravate bradykinesia, tremor, or rigidity. No stimulation-induced dyskinesia was observed postoperatively. These parameters did not need any modification for the next 3 months of follow-up. 

Of note, two months later, a successful transcatheter aortic valve implantation was performed to treat the aortic stenosis as well.

## 4. Discussion

The cardiac pacemakers that are properly programmed and work in a not disturbed environment significantly improve the quality of life of patients with some arrhythmias. There are other devices (high-voltage machines/wires, wave transmitters, other generators), generating electromagnetic waves, which may interfere with normal CIED functioning, however, the patients are clearly informed what to avoid and how to behave. Nevertheless, there are situations in which a cardiac pacemaker has to coexist permanently with another pulse generator, and this is such a case. 

This report shows that the concurrent use of a leadless AV pacemaker with bilaterally implanted DBS is possible and does not have to disrupt their proper functioning. This proceeding allowed us to obtain an accelerometer-based atrioventricular synchronous cardiac pacing and to maintain a relief-bringing optimal DBS. Bongiorni et al. documented a safety coexistence of a single-chamber leadless cardiac pacemaker and DBS [6], but this might be a convenient option mainly for patients with permanent atrial fibrillation. To our best knowledge, this is the first report on a successful leadless AV pacemaker implantation in a patient with bilateral DBS. 

It should be emphasized that there are recommendations issued by the manufacturers of the generators, which should be followed to minimize the risk of potential interference. It is very helpful to have a multidisciplinary team with input from a neurologist/neurosurgeon during the procedure, ready to reprogram the devices adequately if required. It is reasonable to set all devices into a bipolar configuration to limit the size of the effective electrical circuit, which significantly reduces the risk of unfavorable interference. However, the challenge might be to position all generators at an adequate distance one from each other, in particular, if DBS devices are already implanted bilaterally. There are published few reports showing that CIED and DBS pulse generators do not have to interfere even if the devices are located in close proximity from one another (e.g., the same subclavicular region) [7]. However, this is not a rule, and moreover, a close proximity of the generators may contribute to inadvertent interference also when using a magnet/programming device at a follow-up visit or an electrical diathermy during the generator replacement procedure. It is feasible to use the long intravenous leads or the epicardial ones tunneled to the CIED generator positioned in an abdominal area, but especially the latter is not an optimal option if the condition of the patient is frail. Our report shows that a leadless AV cardiac pacemaker may be in these cases a considerable alternative. 

It should be also underlined that chronic right ventricular pacing is not a physiological option for cardiac stimulation. It is effective, but it promotes electrical and mechanical dyssynchrony potentially leading to pacing-induced cardiomyopathy [8]. Fortunately, this is not a common phenomenon in cardiac electrotherapy, therefore leadless pacemaker implanted inside the right ventricle seemed to be the most optimal option in the patient with preserved left ventricular function, and in this clinical setting.

This is just a single report; therefore the caution has to be exerted when a similar clinical context occurs. More such cases are needed to strengthen the evidence that the described proceeding is safe and effective.

## 5. Conclusions

Our report shows that a leadless AV cardiac pacemaker implanted entirely inside the right ventricle might be a reasonable therapeutic option in patients with atrioventricular conduction disorders and bilaterally implanted DBS devices.

## Figures and Tables

**Figure 1 ijerph-20-00388-f001:**
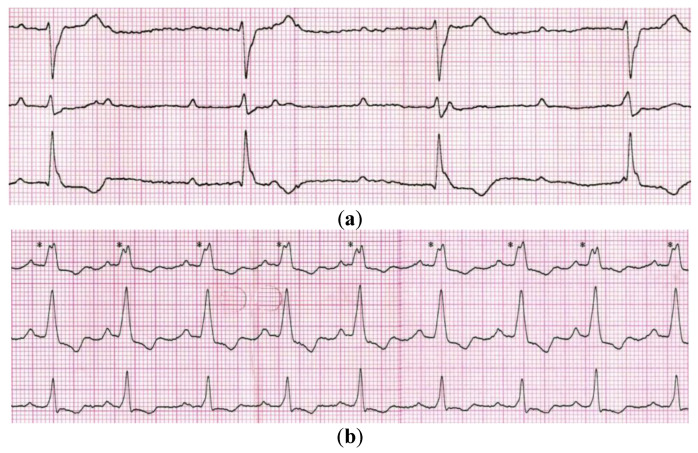
Electrocardiogram showing: (**a**) a sinus rhythm with complete heart block present at the admission time, and (**b**) a sinus rhythm with atrioventricular synchronous cardiac pacing obtained after a leadless AV pacemaker implantation (* - ventricular pacing impulses).

**Figure 2 ijerph-20-00388-f002:**
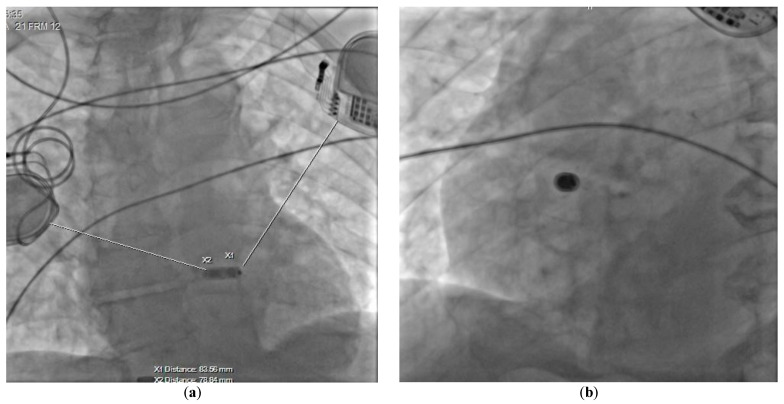
Chest radiogram presenting the final position of the implanted device in (**a**) AP, and (**b**) LAO 32° projections.

## Data Availability

The data supporting the report are available from Marek Szołkiewicz (e.mars@wp.pl) on reasonable request.

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
