# Peer review of "Leadless AV Pacemaker in Patient with Complete Heart Block and Bilaterally Implanted Two Deep Brain Stimulators Can Be Safe Therapeutic Option"

_ijerph, 2022, doi:10.3390/ijerph20010388_

Round 1
Reviewer 1 Report
Dr. Jędrzej Michalik and colleagues presented the case of leadless pacemaker implantation in a patient with bilaterally implanted deep brain stimulation devices. The choice of leadless device seems to be driven by reports of electrical interference between DBS devices and conventional CIEDs using leads.
Comments:
In my opinion, there should be some discussion on whether the potential benefit of a longer distance and thereby lower probability of significant interaction between DBS device and leadless pacemaker implanted in right ventricle sufficiently justifies the known adverse outcomes of chronic right ventricular pacing including mechanical dysfunction leading to heart failure.
Author Response
Thank you very much for a careful analysis of this manuscript. Your note considering the potential negative long-term outcomes of right ventricular pacing is a very accurate one, thus we have addressed this topic in the discussion:
'It should be also underlined that chronic right ventricular pacing is not a physiological option of cardiac stimulation. It is effective, but it promotes an electrical and mechanical dyssynchrony potentially leading to pacing-induced cardiomyopathy. Fortunately, this is not a common phenomenon in cardiac electrotherapy, therefore leadless pacemaker implanted inside the right ventricle seemed to be the most optimal option in this patient, with preserved left ventricular function, and in this clinical setting.'

Reviewer 2 Report
Michalik et al presented a case report which is well written and in interest of the concerned audience.
Author Response
Thank you for the time you devoted to read and to review this manuscript.

Reviewer 3 Report
The authors reported an uncommon case that a patient with bilaterally implanted deep brain stimulation devices, was implanted with a leadless AV pacemaker because of third-degree atrioventricular block. They found that there was no significant interference by DBS on Leadless pace maker. Hence, for these patients with DBS, a leadless AV pacemaker may be a safe and reasonable option for cardiac pacing. This article described the case in detail and no obvious shortage presented. Here, only some minor suggestions were given to improve the manuscript.
1. In line 41, the term “AVA” was showed for the first time, you may present the full name for AVA.
2. Also in line 41, cm2 should be presented as “cm2(top right corner)”.
3. For figure 2, the image of LAO can be showed to us for better understanding “the mid-septum of the right ventricle”.
Author Response
Thank you for taking your time to review this manuscript. We have corrected the indicated mistakes and according to your suggestion, we have added the picture with the LAO 32° projection.
